# The Role of Heme Oxygenase 1 in the Protective Effect of Caloric Restriction against Diabetic Cardiomyopathy

**DOI:** 10.3390/ijms20102427

**Published:** 2019-05-16

**Authors:** Maayan Waldman, Vadim Nudelman, Asher Shainberg, Romy Zemel, Ran Kornwoski, Dan Aravot, Stephen J. Peterson, Michael Arad, Edith Hochhauser

**Affiliations:** 1Cardiac Research Laboratory, Felsenstein Medical Research Institute Petah-Tikva, Sackler Faculty of Medicine, Tel Aviv University, Tel Aviv 49100, Israel; maayanw@gmail.com (M.W.); vadimnud@post.tau.ac.il (V.N.); zemel@post.tau.ac.il (R.Z.); rkornowski@clalit.org.il (R.K.); aravot_dan@clalit.org.il (D.A.); 2Levied Heart Center, Sheba Medical Center, Tel Hashomer and Sackler School of Medicine, Tel Aviv University, Ramat Gan 52621, Israel; michael.arad@sheba.health.gov.il; 3Faculty of Life Sciences, Bar Ilan University, Ramat Gan 5290002, Israel; shaina@mail.biu.ac.il; 4Department of Medicine, New York Presbyterian Brooklyn Methodist Hospital, Brooklyn, NY 11215, USA; 5Department of Medicine, Weill Cornell Medicine, New York, NY 14853, USA

**Keywords:** caloric restriction, Sirtuin 1, Heme Oxygenase-1, PGC-1α, cardiomyopathy, diabetes mellitus

## Abstract

Type 2 diabetes mellitus (DM2) leads to cardiomyopathy characterized by cardiomyocyte hypertrophy, followed by mitochondrial dysfunction and interstitial fibrosis, all of which are exacerbated by angiotensin II (AT). SIRT1 and its transcriptional coactivator target PGC-1α (peroxisome proliferator-activated receptor-γ coactivator), and heme oxygenase-1 (HO-1) modulates mitochondrial biogenesis and antioxidant protection. We have previously shown the beneficial effect of caloric restriction (CR) on diabetic cardiomyopathy through intracellular signaling pathways involving the SIRT1–PGC-1α axis. In the current study, we examined the role of HO-1 in diabetic cardiomyopathy in mice subjected to CR. Methods: Cardiomyopathy was induced in obese diabetic (*db/db*) mice by AT infusion. Mice were either fed ad libitum or subjected to CR. In an in vitro study, the reactive oxygen species (ROS) level was determined in cardiomyocytes exposed to different glucose levels (7.5–33 mM). We examined the effects of Sn(tin)-mesoporphyrin (SnMP), which is an inhibitor of HO activity, the HO-1 inducer cobalt protoporphyrin (CoPP), and the SIRT1 inhibitor (EX-527) on diabetic cardiomyopathy. Results: Diabetic mice had low levels of HO-1 and elevated levels of the oxidative marker malondialdehyde (MDA). CR attenuated left ventricular hypertrophy (LVH), increased HO-1 levels, and decreased MDA levels. SnMP abolished the protective effects of CR and caused pronounced LVH and cardiac metabolic dysfunction represented by suppressed levels of adiponectin, SIRT1, PPARγ, PGC-1α, and increased MDA. High glucose (33 mM) increased ROS in cultured cardiomyocytes, while SnMP reduced SIRT1, PGC-1α levels, and HO activity. Similarly, SIRT1 inhibition led to a reduction in PGC-1α and HO-1 levels. CoPP increased HO-1 protein levels and activity, SIRT1, and PGC-1α levels, and decreased ROS production, suggesting a positive feedback between SIRT1 and HO-1. Conclusion: These results establish a link between SIRT1, PGC-1α, and HO-1 signaling that leads to the attenuation of ROS production and diabetic cardiomyopathy. CoPP mimicked the beneficial effect of CR, while SnMP increased oxidative stress, aggravating cardiac hypertrophy. The data suggest that increasing HO-1 levels constitutes a novel therapeutic approach to protect the diabetic heart. Brief Summary: CR attenuates cardiomyopathy, and increases HO-1, SIRT activity, and PGC-1α protein levels in diabetic mice. High glucose reduces adiponectin, SIRT1, PGC1-1α, and HO-1 levels in cardiomyocytes, resulting in oxidative stress. The pharmacological activation of HO-1 activity mimics the effect of CR, while SnMP increased oxidative stress and cardiac hypertrophy. These data suggest the critical role of HO-1 in protecting the diabetic heart.

## 1. Introduction

Diabetes mellitus type 2 (DM2) is associated with excess cardiovascular morbidity and mortality [1,2]. Diastolic dysfunction, reduced myocardial contractility and heart failure are evident as a result of progressive cardiac fibrosis and/or pressure overload [3,4]. Insulin resistance and hyperinsulinemia, hyperglycemia, and elevated free fatty acids are primary factors that lead to cardiomyocyte injury, dysfunction and myocardial lipotoxicity in diabetes [5]. Oxidative stress, mitochondrial dysfunction, abnormal intracellular calcium metabolism [6] and chronic inflammation [7] are mediators of cardiac damage. Angiotensin II (AT) is a potent vasoconstrictor [8]. Endogenous cardiac AT synthesis triggers the development of cardiac hypertrophy [9] irrespective of hypertension [10].

Individuals voluntarily practicing long-term Caloric Restriction (CR) suggest that it favorably affects cardiovascular disease risk factors [11], simultaneously postponing age-related diseases and longevity in animal models [12,13]. Adiponectin, which increases in the plasma after CR [14,15], has been implicated in CR-induced cardioprotection [14]. Sirtuin-1 (SIRT1), a redox-sensitive enzyme, is a member of a large family of class III histone deacetylases (HDAC) [16,17]. It modulates genetic stability, extending the life span of flies, and worms [18]. SIRT1 regulates cellular processes such as apoptosis/cell survival, chromatin remodeling, and gene transcription [19]. SIRT1 activation by CR drives a number of downstream events, including the peroxisome proliferator-activated receptor gamma coactivator 1-alpha (PGC-1α) and the anti-oxidant protein heme oxygenase (HO-1) [20,21]. There are decreased cardiac levels of HO-1 and adiponectin and elevated levels of inflammatory cytokines (Tumor Necrosis Factor α: TNFα) and the oxidative stress marker malondialdehyde (MDA) in the sera of diabetic patients [20]. AT release by the adipocyte and the reduction of HO-1 lead to reactive oxygen species (ROS) and oxidative stress. These factors have a decisive role in obesity-induced injury, mitochondrial dysfunction, and fragmentation [22,23]. Together, these proteins improve metabolic signaling pathways, and blunt pro-inflammatory pathways in mice fed a high-fat, high-calorie diet [24,25]. ROS dependent perturbations associated with metabolic syndrome are influenced by HO activity [26,27]. It is suggested that the genes associated with lipid metabolism, adipocyte differentiation and insulin sensitivity upregulation are influenced by the nuclear transcription factors, peroxisome proliferator-activated receptors (PPARs), i.e., PPARα, γ, and δ [28,29].

Using a murine cardiomyopathy model obtained by stressing the diabetic heart by AT, we reported that CR decreased cardiac hypertrophy and inflammatory markers [30,31]. We also showed that CR affects cardiac remodeling in these mice through molecular mechanisms related to mitochondrial function and an antioxidative signaling pathway mediated by SIRT1 and PGC-1α [31,32]. In the present study, using the same model, we identify HO-1 as a key factor behind the cardioprotective effect of CR. We demonstrate that increased levels of HO-1 improve antioxidant defense and enhance metabolic adaptation through its interaction with SIRT1. Increasing HO-1 mimicked the protective effects of CR on the diabetic heart, while the inhibition of HO activity increased oxidative stress and aggravated pathological hypertrophy.

## 2. Results

### 2.1. CR Reduced Oxidative Stress and Increased PGC-1α and HO-1 Levels

We have previously characterized the murine model of diabetic cardiomyopathy by combining diabetes (*db/db* transgenic mice) and AT infusion, and reported the cardioprotective effects of CR [31]. The model of cardiomyopathy in AT-stressed diabetic mice and the protective effect of CR are described in Table 1. Body weight, glucose, Aspartate Aminotransferase (AST), Alanine aminotransferase (ALT), and cholesterol triglycerides were all higher in diabetic mice compared to WT mice. AT induced cardiomyopathy, as demonstrated by both functional and biochemical markers. In order to examine the role of HO-1 in the cardioprotection afforded by CR, SnMP was administrated to the diabetic mice concomitantly with CR. SnMP resulted in increased levels of AST, GOT, and of cholesterol, reversing the beneficial effects of CR. AT with and without diabetes reduced HO-1 levels of cardiac tissue compared to non-treated WT animals, (*p* = 0.001), while CR increased HO-1 (*p* = 0.02, Figure 1). MDA levels were increased in *db/db* + AT mice compared to WT mice (*p* = 0.01), but fell following CR (*p* < 0.03) (Figure 2A).

CR had a beneficial metabolic effect on blood lipids, but that was abolished by SnMP (cholesterol; *p* = 0.04, triglycerides; *p* = 0.006) with no significant effect on both body weight and blood glucose. SnMP resulted in left ventricular hypertrophy (LVH), preventing the protective effect of CR on cardiac hypertrophy (*p* = 0.003). SnMP also increased systolic blood pressure (BP) to the level found in diabetic AT-treated mice without CR (*p* = 0.005) (Table 1), and increased MDA levels (Figure 2A). Adiponectin was reduced in diabetic mice, while AT and CR-treated animals displayed elevated adiponectin levels and SIRT1 activity, which was blocked by SnMP (Figure 2B,C). PGC-1α was reduced in diabetic AT-treated heart tissue (*p* < 0.001). PGC-1α levels were elevated following CR (*p* < 0.0001), but reduced following SnMP treatment (Figure 2D). PPARγ levels were higher in diabetic mice compared to WT. CR reduced PPARγ levels *db/db* +AT hearts. SnMP abolished the beneficial effects of CR, reducing the levels of adiponectin, PGC-1α, and SIRT1 to those of diabetic mice (Figure 2B–D), while increasing PPARγ levels (Figure 2E).

### 2.2. Cross-Talk between HO-1-SIRT1-and PGC-1α

In order to examine the interaction between HO-1–SIRT1–PGC-1α and their role in glucose metabolism and oxidative stress in the heart, cultured rat neonatal cardiomyocytes exposed to different concentrations of glucose (7.5 mM, 17.5 mM, and 33 mM) were used. Elevated glucose levels led to a concomitant increase in cellular ROS production (*p* < 0.03) (Figure 3(Aa,d,g,B)) and reduction in SIRT1 and PGC-1α proteins levels (*p* < 0.05, Figure 3A,C–F). HO-1 inhibitor SnMP produced a significant reduction in the levels of both SIRT1 (*p* < 0.002) and PGC-1α (*p* < 0.009) (Figure 3C–F), leading to increased ROS production (*p* < 0.001, Figure 4Ab,e,h). The HO-1 inducer, CoPP, increased SIRT1 (*p* < 0.008) and PGC-1α expression (*p* < 0.03, Figure 3D–F) and prevented the glucose-mediated elevation of ROS (*p* < 0.002, Figure 3Ac,f,i). As shown in Figure 3G, the basal levels of HO activity is inhibited by about 70% in bilirubin formation in the presence of SnMP. An increase of glucose levels caused the inhibition of HO activity, and was further potentiated by SnMP, which is clearly observed when glucose reached 33 mM (* *p* < 0.01 vs. control, # *p* < 0.001 vs. SnMP).

SIRT1 inhibition by EX-527 elevated ROS production (Figure 4A,B). PGC-1α and HO-1 protein levels also decreased (Figure 4C–E). Cumulatively, these results indicate a direct bilateral relationship between SIRT1–PGC-1α and HO-1; perturbations in HO activity and HO-1 levels influence upstream molecules e.g., SIRT1. Therefore, SIRT1–PGC-1α and HO-1 form a pathway with a positive feedback loop protecting cardiomyocytes against oxidative stress, which participates in the pathogenesis of diabetic heart disease (Figure 5).

## 3. Discussion

Obesity affects major segments of the population. We have previously shown that oxidative stress is implicated in the pathogenesis of insulin resistance and its consequent vascular injury. We have emphasized the role of reactive oxygen species in adipocytes that resulted in decreased adiponectin levels, increased inflammation, and decreased adipogenesis. In this study, we identified that the CR-mediated cardio protection effect is dependent on HO-1 expression. Furthermore, we showed a direct link between HO-1-SIRT1 and PGC-1α signaling and the attenuation of diabetic cardiomyopathy.

There are numerous pathways involved in different models of cardiomyopathy [33]. Cardiac hypertrophy is mediated in part by the RAS and TGF-β, which have a central role in cardiac remodeling. Since four-month-old diabetic mice did not develop cardiac hypertrophy or fibrosis, we developed an AT-dependent murine cardiomyopathy model by further stressing the diabetic heart by AT [31]. AT has been reported to induce cardiomyopathy mainly through its profibrotic effects [2,10]. Oxidative stress plays a pivotal role in the development of obesity and the pathogenesis of diabetes [6]. In the current study, SnMP led to a marked hypertrophic remodeling in excess of that present in AT-treated *db/db* mice. Perturbations in adiponectin and PPARγ that are closely related to HO-1 are also involved in the hypertrophic remodeling as well [34,35]. While oxidative stress decreases adiponectin, HO-1 helps increase adiponectin, thereby preventing cardiomyopathy and heart failure development [36]. AT with and without diabetes reduced HO-1 levels of cardiac tissue compared to non-treated WT animals, while CR increased HO-1. MDA levels were increased in *db/db* + AT mice compared to WT mice, but fell following CR, demonstrating the antioxidative role of HO-1 in cardiac tissues.

PGC-1α has been characterized as a master regulator of mitochondrial biogenesis. It acts through several transcription factors, including Nuclear Respiratory Factor (NRF1 and NRF2), which regulate the expression of antioxidant genes, including HO-1 [37]. We have previously shown that the activation of PGC-1α reduced mitochondrial ROS in adipocytes through the induction of HO-1, and that the silencing of PGC-1α prevented the increased levels of HO-1 in these cells [38]. PGC-1α is not activated until it is deacetylated by SIRT1 [39,40], thereby helping antioxidant defenses [41]. The levels of mitochondrial cofactors SIRT1, PGC-1α, and HO-1 were reduced in diabetic AT-treated hearts, while CR elevated these factors. On the contrary, SnMP produced a significant reduction in the levels of both SIRT1 and PGC-1α, leading to increased ROS production. These results establish the link between SIRT1, PGC-1α, and HO-1 signaling that leads to the attenuation of ROS production and diabetic cardiomyopathy.

The downregulation of SIRT1 has been implicated as a contributing factor in metabolic disorders, inducing the metabolic syndrome and DM2 [42]. The SIRT1 protein binds to and represses genes controlled by the fat regulator PPARγ [43]. During CR, fatty acids levels are reduced in diabetic mice, resulting in reduced lipotoxicity [30,44]. CR reduced PPARγ levels, consequently preventing the initiation of the cascade that leads to lipotoxicity that participates in the cardiomyopathy process.

Heme oxygenase exists in two forms, HO-1(inducible) and HO-2 (non-inducible), and is rate limiting in heme degradation to biliverdin, iron, and carbon monoxide. Biliverdin is rapidly converted to bilirubin with positive effects on numerous biological functions [27]. The pleiotropic effects of HO-1 on obesity and cardiovascular disease is well documented [26,45]. HO-1 exhibits a broad spectrum of actions on blood vessel endothelial tissue. This includes increased levels of vasodilation, increased numbers of endothelial progenitor cells, and improved cardiac cell function, while decreasing vasoconstriction and inflammation [26,46]. Increased levels of HO-1 result in increased levels of the antioxidant, bilirubin, and the antiapoptotic, carbon monoxide, which are responsible for neutralizing free radicals, ICAM-1, VCAM-1, TNF, and IL-18 [27]. The role of inflammation and HO-1 in cardiac diabetes using the strepotozocin (STZ) model has been previously described. Myocardial fibrosis and apoptosis, but not inflammation, were found in long-term experimental diabetes STZ [47]. Previously published reports showed that HO-1 induction attenuates glucose-mediated cell growth arrest and apoptosis in human and mice cell line [48]. Additionally, high levels of glucose and hyperglycemia inhibits HO-1 activity and expression, as glucose deprivation increases HO-1 expression [49,50]. CR offers effective protection on later responses such as hypertrophy. Thus, the upregulation of HO-1 in CR offers cytoprotection that is manifested in the amelioration of cardiovascular disease and protection against cardiomyopathy [51,52]. Increased levels of HO-1 through pharmacologic intervention with compounds such as resveratrol and L4F result in cardiac improvement that is akin to that observed with CR. In summary, pharmacologic or genetic interventions to increase HO-1 constitute a novel therapeutic approach to preventing diabetic cardiomyopathy in humans.

In conclusion, in the current study, we show that the cardioprotective effect of CR in diabetic mice involves the increased expression of PGC-1α in association with increased HO-1 and SIRT1 levels. The dependency of the cell survival on HO activity was evident (Figure 3G). Thus, the mechanism of the cell protection in glucose is partially dependent on HO activity. The inhibition of HO activity by SnMP abolished the beneficial effect on cardiac metabolic dysfunction represented by adiponectin, SIRT1, PPARγ, PGC-1α, MDA, and pathological cardiac hypertrophy. The pharmacological inhibition of either HO-1 or SIRT1 of isolated cardiomyocytes was followed by the decreased expression of SIRT1 and PGC-1α and an elevation of PPARγ levels. In contrast, CoPP increased the levels of SIRT1, PGC-1α, and HO-1, and attenuated the myocardial RO, suggesting a mutual symbiotic relationship between these cardioprotective mediators. Prior studies demonstrated that increased levels of HO have been shown to attenuate the expression of inflammatory markers through a number of mechanisms [53,54,55]. The current data suggest that the increased expression of SIRT1 and PGC-1α is responsible for the increased levels of HO-1. This may be considered as a pivotal axis that is the first line of defense against oxidative stress caused by hyperglycemia, and is essential to protect the diabetic heart from insults. While the field of pharmacological therapies continues to expand, efforts to facilitate weight loss have had limited success. In the present study, we examined the cellular mechanism by which CR protects the diabetic heart. We must understand the underlying cellular mechanisms in order to prevent adverse cardiac remodeling. Our findings are crucial for the development of novel therapeutic approaches such as targeting the HO-1–SIRT1–PGC-1α axis to prevent cardiomyopathy and heart failure, which is a major source of morbidity and mortality in diabetic patients.

## 4. Materials and Methods

### 4.1. Animal Model

The animal experiments were approved by the institutional animal care and use committee of Tel Aviv University (M-15-010, 16 February 2015). Homozygous *db/db* mice (C57BLKS/J-*leprdb/leprdb*) and their wild-type (WT) littermates were maintained in a pathogen-free facility on regular rodent chow with free access to water and 12-h light and dark cycles. Homozygous mice were verified by PCR. Male WT or *db/db* mice (12–14 weeks old) were used for the experiments. *db/db* mice develop mild cardiomyopathy at an advanced age [2,56]. To enhance the development of heart disease and obtain a robust phenotype, mice were stressed by ATII as described in other cardiomyopathy models [10]. Mice were divided into the following groups, *n* = 5–14 each in each group: WT, *db/db, db/db* + AT, *db/db* + AT + CR, and *db/db* + AT + CR + SnMP.

### 4.2. Angiotensin

Mice were anesthetized with 2% isoflurane, and an ALZET osmotic pump (Durect Corp., Cupertino, CA, USA) was subcutaneously implanted into each mouse. The osmotic pumps infused angiotensin II (Sigma-Aldrich, St. Louis, MO, USA) at a rate of 1000 ng·kg^−1^·min^−1^ for 4 weeks.

### 4.3. Caloric Restriction

Mice were housed in individual cages. Caloric-restricted (CR) mice were fed 90% of their average caloric intake for 2 weeks (10% restriction), followed by 65% of that for an additional 2 weeks (35% restriction). Experiments were conducted after the 4-week period, as we have previously described [57].

### 4.4. Cell Culture

Rat hearts (Sprague–Dawley 1–2 days old) were removed under sterile conditions and washed three times in phosphate-buffered saline (PBS) to remove excess blood cells. We used rat culture because the rat heart is bigger than the mouse heart; therefore, the yield of cardiomyocytes is higher. The hearts were minced and then gently agitated in a solution of proteolytic enzymes, RDB (Biological Institute, Ness-Ziona, Israel), which was prepared from fig tree extract. RDB was diluted 1:100 in Ca^2+^ and Mg^2+^-free PBS for a few cycles of 10 min each, as previously described [58]. Dulbecco’s modified Eagle’s medium (Biological Industries, Kibbutz Beit Haemek, Israel) containing 10% horse serum was added to supernatant suspensions containing dissociated cells. The mixture was centrifuged at 300 g for 5 min. The supernatant was discarded, and the cells were resuspended. The suspension of the cells was diluted to 1.06 × 10^6^ cells/mL, and 1.5 mL of the suspension was placed in 35-mm plastic culture dishes, or 0.5 mL in 24-well plates. The cultures were incubated in a humidified atmosphere of 5% CO_2_ and 95% air at 37 °C. Confluent monolayers exhibiting spontaneous contractions developed in culture within 2 days [31].

### 4.5. Experiments with EX-527, CoPP, SnMP, and HO Activity

Cultured cardiomyocytes were incubated with different concentration of glucose (7.5 mM, 17.5 mM, and 33 mM) for 4 days. A glucose concentration of 17.5 mM was considered as control. The SIRT1 inhibitor EX-527 (Cayman Chemical, Ann Arbor MI, USA) was added to the culture for 24 h (10 µM). Cobalt protoporphyrin dichloride (CoPP), 2 µM, which increases HO-1 protein levels, and HO activity and Sn(tin)-mesoporphyrin dichloride (SnMP) 1 µM, inhibits HO activity (Frontier Science, Logan, UT, USA), were dissolved in 0.1 M of sodium citrate buffer, pH 7.8 and added to the cardiomyocyte cultures for 72 h [59]. For the in vivo study, SnMP (2 mg/100 g, intraperitoneal was injected every 4 days concomitantly with AT infusion and CR. HO activity was measured by incubating myocyte in the presence of glucose using the same methods by Da-Silva et al. [60] in which bilirubin, the end product of heme degradation by HO, was extracted with chloroform, and its concentration was determined spectrophotometrically (Dual UV/VIS Beam Spectrophotometer Lambda 25; Perkin-Elmer, Norwalk, CT, USA) using the difference in absorbance at wavelength from λ 460 to λ 530 nm with an absorption coefficient of 40 mmol/L^−1^ and cm^−1^.

### 4.6. In Vitro ROS Production Measurement

ROS was detected using a 2′, 7′-dichlorofluorescin diacetate (DCF-DA) reagent (Sigma-Aldrich, St. Louis, MO, USA). This compound is an uncharged cell-permeable molecule. Inside cells, this probe is cleaved by non-specific esterases, forming carboxy dichlorofluoroscein, which is oxidized in the presence of ROS. Cells were loaded with 10 µM of DCF-DA for 30 min at 37 °C and then washed. Fluorescence was monitored with a microplate fluorimeter using wavelengths of 485/538 nm for excitation/emission, respectively.

### 4.7. Western Blotting

Cardiac tissue was homogenized in lysis buffer and quantified for protein levels using a commercial assay (Bio-Rad, Hercules, CA, USA). Western blotting was performed according to standard procedures, as previously described [31,61]. Protein samples (60 μg) were applied to sodium dodecyl sulfate (SDS) polyacrylamide gel (10–15%), electrophoresed under denaturing conditions and electrotransferred onto nitrocellulose membranes (Bio-Rad). Membranes were blocked with 3% BSA in tris-buffered saline (TBS). Primary antibodies for β actin, GAPDH (Santa Cruz Biotechnology, Dallas, Texas, USA), PGC-1α (ABCAM, Cambridge, UK), HO-1 (Enzo Life Sciences, Farmingdale, NY, USA), and SIRT1 (Merck Millipore Corp, Darmstadt, Germany) were used in TBST with 3% BSA overnight at 4 °C. Dye 680 or 800 secondary antibodies were added at a concentration of 1:10,000 for 1 h at room temperature (LI-COR Biosciences, Lincoln, NE, USA). Detection was carried out with the LI COR Odyssey. Quantification of signals was carried out with the Odyssey program. The ratio between the intensity of the band of the tested protein and the intensity of the corresponding actin or GAPDH band was calculated for the normalization/expression of results.

### 4.8. RT-PCR

Total RNA was purified from hearts using TRIzol (Ambion, Austin, TX, USA) as per the manufacturer’s instructions. The quantity of total RNA was determined by OD260 measurements. cDNA was synthesized from total RNA using the TaqMan High Capacity cDNA Reverse Transcription Kit (Applied Biosystems, Foster City, CA, USA) according to the manufacturer’s protocol. Quantitative real-time PCR analysis was performed using the Step One Plus system (Applied Biosystems, Foster City, CA, USA). The primers and TaqMan FAM probes were ordered from Applied Biosystems [31].


**Gene**

**Assay ID**
Tbp (TATA BOX)Mm00446973Ppargc1 (PGC-1α)Mm01208835Adipoq (adiponectin)Mm00456425

### 4.9. Serum Thiobarbituric Acid Reactive Substances

Malondialdehyde (MDA) was quantified through a controlled reaction with thiobarbituric acid, generating thiobarbituric acid-reactive substances (TBARS). Thus, lipid peroxidation was determined using the TBARS assay kit (Cayman Chemical, Ann Arbor, MI, USA) according to the manufacturer’s instructions.

### 4.10. SIRT Activity

SIRT activity in the nuclear fraction of cardiac tissue samples was measured using the Universal SIRT activity assay kit (Abcam, Cambridge, UK).

### 4.11. Statistical Analysis

Animals were assigned to groups randomly. All the values were expressed as mean ± SD. In the in vivo studies, results were normalized to the WT group, and in the in vitro studies, the results were normalized to the 17.5-mM glucose control group. The statistical difference between the two groups was assessed using the two-tailed Student’s *t*-test. To compare more than two groups, one-way analysis of variance (ANOVA) with Duncan’s multiple comparison option was used. *p* < 0.05 was considered significant.

## Figures and Tables

**Figure 1 ijms-20-02427-f001:**
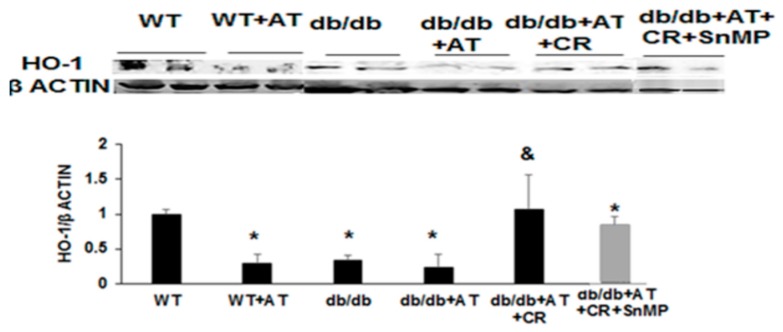
Cardiac heme oxygenase-1 (HO-1) proteins levels: caloric restriction (CR) alleviates oxidative stress through the activation of HO-1. HO-1 was reduced after angiotensin II (AT) treatment in cardiac tissue both in wild-type (WT) and diabetic mice compared to non-treated WT mice (*p* = 0.001), but was elevated after CR. *n* = 4 in each group, * *p* < 0.05 vs. WT, & *p* < 0.05 vs. *db/db* + AT. Values represent mean ± SD.

**Figure 2 ijms-20-02427-f002:**
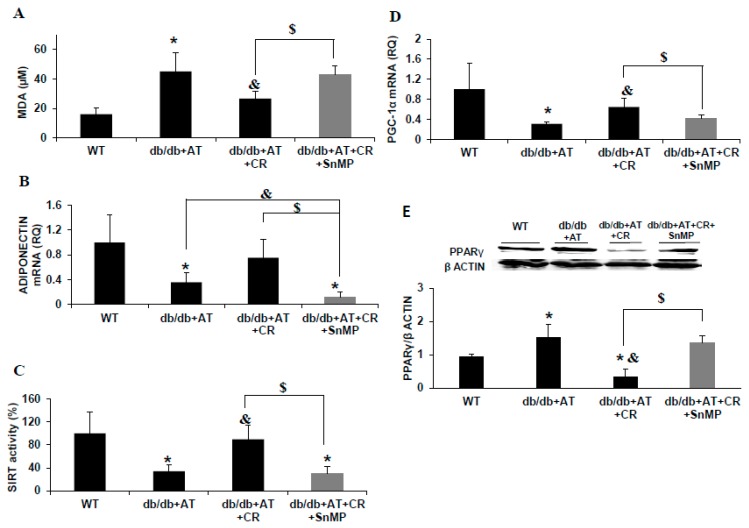
Sn(tin)-mesoporphyrin (SnMP) prevents the beneficial cellular effect of CR. CR diabetic mice were concomitantly treated with SnMP. Malondialdehyde (MDA) levels in the serum were measured using thiobarbituric acid-reactive substances (TBARS) kit, *n* = 4 in each group. *n* = 4 in each group, * *p* < 0.007 vs. WT, & *p* = 0.009 vs. *db/db* + AT, $ *p* < 0.005 vs. *db/db* + AT + CR. Values represent mean ± SD (**A**). Adiponectin (**B**), SIRT1 (**C**), and peroxisome proliferator-activated receptor-γ coactivator (PGC-1α) (**D**) mRNA levels were measured in the cardiac tissue. *n* = 4 in each group, * *p* < 0.04 vs. WT, & *p* < 0.05 vs. *db/db* + AT, $ *p* < 0.04 vs. *db/db* + AT + CR. Values represent mean ± SD. Western blot for peroxisome proliferator-activated receptor y (PPARγ) protein and densitometry analysis of PPARγ normalized to β actin. *n* = 4 in each group, * *p* < 0.03 vs. WT, & *p* = 0.004 vs. *db/db* + AT, $ *p* = 0.002 vs. *db/db* + AT + CR. Values represent mean ± SD (**E**). HO-1 protein levels were reduced in the heart (**F**).

**Figure 3 ijms-20-02427-f003:**
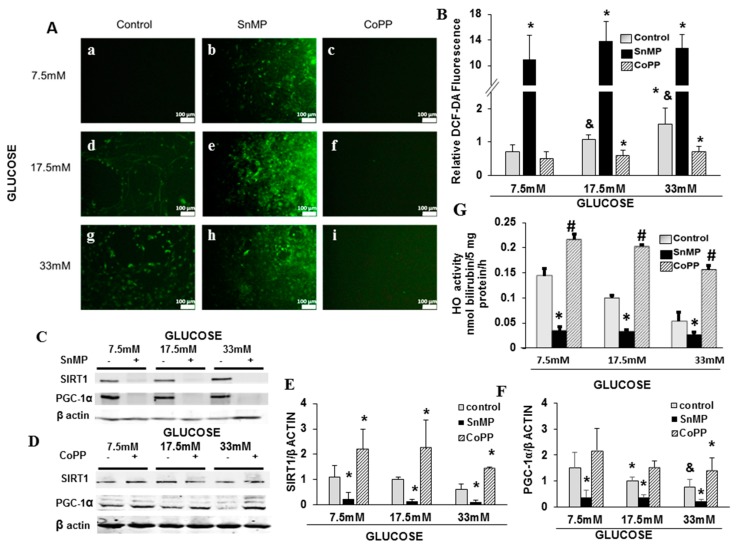
HO-1 is required for the expression of SIRT1 and PGC-1α. Neonatal rat cardiomyocytes were exposed to 7.5 mM, 17.5 mM, and 33 mM of glucose and treated with SnMP or cobalt protoporphyrin (CoPP). Cells were stained with 2′, 7′-dichlorofluorescin diacetate (DCF-DA) (**A**a–i), and fluorescence was measured using a fluorimeter (**B**). Representative western blots for SIRT1 and PGC-1α for cells treated with SnMP (**C**) and CoPP (**D**), densitometry analysis for SIRT1 (**E**) and PGC-1α (**F**). HO activity in the presence and absence of SnMP and CoPP (**G**). Results were normalized to the group exposed to 17.5 mM of glucose. * *p* < 0.05 vs. control, & *p* < 0.05 vs. 7.5 mM of control, # *p* < 0.05 vs. SnMP. *n* = 4 in each group. Values represent mean ± SD.

**Figure 4 ijms-20-02427-f004:**
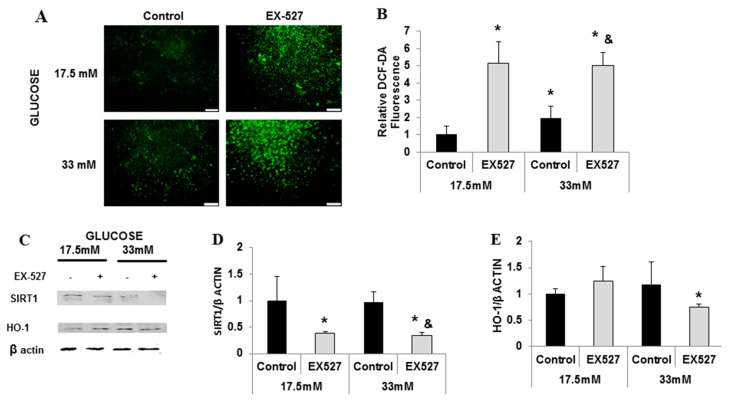
SIRT1 is required for the expression of PGC-1α and HO-1. Neonatal rat cardiomyocytes were exposed to 17.5 mM and 33 mM of glucose and treated with the SIRT1 inhibitor EX-527. Cells were stained with DCF-DA, Scale bar: 100 μm. (**A**a–d), and fluorescence was measured using fluorimeter (**B**). Representative Western blots for SIRT1, PGC-1α, and HO-1 for cells treated with EX-527 (**C**), densitometry analysis for SIRT1 (**D**), and HO-1 (**E**). Results were normalized to the group exposed to 17.5 mM of glucose. * *p* < 0.05 vs. control, & *p* < 0.05 vs. 33 mM of control, *n* = 4 in each group. Values represent mean ± SD.

**Figure 5 ijms-20-02427-f005:**
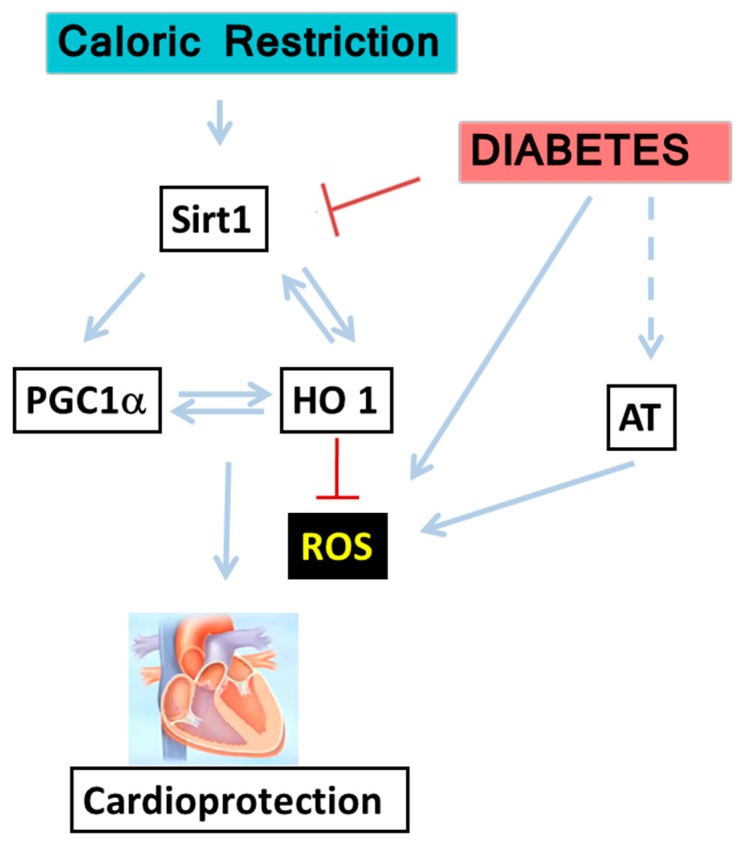
Illustration of CR cellular signaling hereby suggested to participate in the development of Type II diabetic cardiomyopathy: The energetic dysfunction in diabetes increase in the heart together with elevation in the production of angiotensin, leading to mitochondrial dysfunction, oxidative stress, and inflammation. CR elevates adiponectin and SIRT-1 levels. These lead to the activation of both PGC-1α and HO-1, which together improve mitochondrial function, alleviate the oxidative stress, and reduce inflammation by CR, ameliorating cardiomyopathy. SIRT-1, PGC-1α, and HO-1 form a positive feedback loop elevating each other and thus protecting cardiomyocytes against oxidative stress, which participates in the pathogenesis of diabetic heart disease.

**Table 1 ijms-20-02427-t001:** The effect of CR on LV dimension and biochemistry.

	WT *n* = 8	*db/db* *n* = 14	*db/db* + AT *n* = 14	*db/db* + AT + CR *n* = 8	*db/db*+AT+ CR + SnMP *n* = 5
IVS (mm)	0.8 ± 0.1	0.9 ± 0.1	1.1 ± 0.1 #	1 ± 0.1 ^&^	1.3 ± 0.1 ^$^
LVPW (mm)	0.9 ± 0.1	0.9 ± 0.1	1.1 ± 0.2 #	0.9 ± 0.2 ^&^	1.3 ± 0.3 ^$^
LVEDD (mm)	3.6 ± 0.7	3.9 ± 0.2	3.5 ± 0.05 #	4.1 ± 0.4 ^&^	3 ± 1.2 ^$^
LVESD (mm)	2.9 ± 0.2	2.6 ± 0.3	2.4 ± 0.6	2.5 ± 0.5	2.1 ± 0.4
FS (%)	33 ± 14	34 ± 7	34 ± 7	41 ± 10 ^&^	40 ± 4 ^&^
Body Weight (g)	26 ± 3	41 ± 10 *	40 ± 5	33 ± 7 ^&^	36 ± 6
Systolic Blood Pressure (mmHg)	95 ± 21	99 ± 30	148 ± 15 #	114 ± 11 ^&^	138 ± 9 ^$^
Glucose (mg/dL)	137 ± 44	617 ± 93 *	658 ± 107	531 ± 127 ^&^	427 ± 195 ^&^
AST (U/L)	62 ± 25	127 ± 53 *	226 ± 149	99 ± 21 ^&^	138 ± 122
ALT (U/L)	126 ± 42	182 ± 134	281 ± 176	117 ± 32 ^&^	194 ± 198
Cholesterol (mg/dL)	79 ± 24	112 ± 21 *	199 ± 91 #	118 ± 25 ^&^	156 ± 29 ^$^
Triglycerides (mg/dL)	124 ± 57	185 ± 66 *	208 ± 75	127 ± 35 ^&^	188 ± 19 ^$^

Values are mean ± SD. * *p* < 0.05 vs. WT, # *p* < 0.05 vs. *db/db,*
^&^
*p* < 0.05 vs. *db/db* + AT, ^$^
*p* < 0.05 vs. *db/db* + AT + CR. IVS, intra ventricular septum; LVPW, left ventricle posterior wall; LVESD, left ventricle end systolic dimension; LVEDD, Left ventricle end diastolic dimension; FS, Fractional shortening.

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
