# Peer review of "The Role of Heme Oxygenase 1 in the Protective Effect of Caloric Restriction against Diabetic Cardiomyopathy"

_ijms, 2019, doi:10.3390/ijms20102427_

Round 1
Reviewer 1 Report
Title: The critical role of Heme Oxygenase 1 in the protective effect of caloric restriction against diabetic cardiomyopathy
Ms no. ijms-459926
This research paper describes the central role of HO-1 as a key factor for the protection from mitochondrial dysfunction induced by cardiomyopathy provoked by DM2.
English should be improved.
Main concerns:
1. Although the experiments seems well conceived, several confusing data are presented.
First of all, figure 1 shows 6 different mice treatments, whereas in the following table 1 and figure 2 some treatments are missing.
2. Since these data are HO-1-centered, it would be necessary to measure HO-1 enzymatic activity at least in the in vitro model.
3. Figure 5 resumes main findings. In my opinion is not clear if authors proved the interplay between HO-1 and PGC1alpha; moreover since EX527 (sirt-1 inhibitor) decreases HO-1 level, an arrow from sirt-1 to HO-1 should be added.
4. Finally, it is claimed that the final result is an increase in mitochondrial function, therefore I suggest to measure some specific mitochondrial markers.
Minor concerns:
Please fill-in Authors contributions, Funding, Acknowledgments and Conflict of Interests sections.
Author Response
Title: The critical role of Heme Oxygenase 1 in the protective effect of caloric restriction against diabetic cardiomyopathy
Reviewer1:
1. Although the experiments seem well conceived, several confusing data are presented.
First of all, figure 1 shows different mice treatments, whereas in the following table 1 and figure 2 some treatments are missing.
In fig 1 we wanted to show cardiac HO-1 proteins levels in control – WT vs. Diabetic mice. In the other parts of the invivo experiments the role of HO-1 in the diabetic mice alleviating oxidative stress through the activation of HO-1 was emphasized.
2. Since these data are HO-1-centered, it would be necessary to measure HO-1 enzymatic activity at least in the in vitro model.
We agree with the reviewer that the enzymatic activity of HO-1 is missing but the fact that in the absence of HO1 either by SIRT or HO-1 inhibition emphasizes crucial role of HO-1 in alleviating oxidative stress.
3. Figure 5 resumes main findings. In my opinion is not clear if authors proved the interplay between HO-1 and PGC1alpha; moreover since EX527 (SIRT-1 inhibitor) decreases HO-1 level, an arrow from SIRT-1 to HO-1 should be added.
An arrow from SIRT-1 to HO-1 was added to the scheme in figure 5.
4. Finally, it is claimed that the final result is an increase in mitochondrial function; therefore I suggest measuring some specific mitochondrial markers.
Mitochondrial function and an anti-oxidative signaling pathway mediated by SIRT1, and PGC-1α was discussed and measured in the introduction (lines89-91) and in the discussion (lines 284-294).
5 .Please fill-in Authors contributions, Funding, Acknowledgments and Conflict of Interests sections.
Authors contributions We added these details to the end of the article: page 11.
Consent for Publication: All authors have seen and approve of the data presented.
Conflict of Interests sections-none
6. The key words were added to page 1.
Reviewer 2 Report
This study examine the mechanism of caloric restriction for protection against diabetic cardiomyopathy. The paper is generally well written but there are points that require clarification.
Diabetic cardiomyopathy is not one thing, it is hypertrophy, fibrosis, apoptosis, etc. The authors show HO1 to play a critical role in this process but where in the continuum of the evolution of this process does it fit?
You can exclude lines 49&50 and start with line 51 ("Diabetic cardiomyopathy.....")
Line 92 - please fix the sentence. Something is missing.
Line 115 - why are you using RAT cultures and not mouse cells?
Lines 222ff - the writing is dense. Can it be simplified?
I wonder if some of the information in lines 294ff should not be in the INTRO to give the reader a sense of why HO is so important.
Author Response
Title: The critical role of Heme Oxygenase 1 in the protective effect of caloric restriction against diabetic cardiomyopathy
Reviewer 2:
1. You can exclude lines 49&50 and start with line 51 ("Diabetic cardiomyopathy.....")
We deleted the first part of the sentence as the reviewer suggestion and left: “Diabetes mellitus type 2 (DM2) is associated with excess cardiovascular morbidity and mortality” (lines 52-53).
2. Line 92 - please fix the sentence. Something is missing
The sentence was corrected to: “Increasing HO1 mimicked the protective effects of CR……” line 94.
3. Line 115 - why are you using RAT cultures and not mouse cells?
We used rat culture because the rat heart is bigger than the mouse heart therefore the yield of cardiomyocytes is higher and the number of newborns is smaller. We added this explanation to the manuscript page 3 lines 119-121.
4. Lines 222ff - the writing is dense. Can it be simplified?
The sentence was simplified: “PGC-1α levels were elevated following CR (P<0.0001), but reduced following SnMP (Figure 2D). PPARγ levels were higher In diabetic mice compared to WT. CR reduced PPARγ levels db/db +AT hearts” page 7 lines 218-222.
5. I wonder if some of the information in lines 294ff should not be in the INTRO to give the reader a sense of why HO is so important.
We chose to leave the information on our previous works concerning PGC-1α activation and reduced mitochondrial ROS in adipocytes through induction of HO-1 and SIRT1 [44, 45] in the discussion to indicate that this cellular mechanism exist in different cells and not just the heart .
Reviewer 3 Report
Authors studied the role of HO-1 in DM2 rats with DCM and caloric restriction. The idea is original and may suggest interesting data for research. However, some issues must be corrected or added - Most of the blots are not representative for the quantification showed in the graphs. Please, review - The quality of some blots is very low. Figures 1, 2 and 4. They look chopped and flatten. Please, change - The data from WT animals should appear also in table 1. Similarly, the data of db/db rats should be shown in Figure 2. - The role of inflammation and HO-1 in cardiac diabetes have been previously described (suggested reference Ares-Carrasco et al. 2009). Thus, this response can be short and early in DCM development, and caloric restriction may be more effective on later responses such as hypertrophy or apoptosis then. - Why number of animals is different among the groups? Why is only n=5 in db/db+AngII+CR+SnMP group? Is there any high mortality associated? - What kind of diet animals (wt and db/db) took? - Is angiotensin II the one that you infused? If so, please change to AngII along the manuscript and group of rats - The final scheme could be improved with the localization of proteins in a cell. Diabetes refers to type-II diabetes
Author Response
Title: The critical role of Heme Oxygenase 1 in the protective effect of caloric restriction against diabetic cardiomyopathy
Reviewer3:
The quality of some blots is very low. Figures 1, 2 and 4. They look chopped and flatten. Please, change - The data from WT animals should appear also in table 1. Similarly, the data of db/db rats should be shown in Figure 2. - The role of inflammation and HO-1 in cardiac diabetes have been previously described. Thus, this response can be short and early in DCM development, and caloric restriction may be more effective on later responses such as hypertrophy or apoptosis then. - Why number of animals is different among the groups? Why is only n=5 in db/db+AngII+CR+SnMP group? Is there any high mortality associated? - What kind of diet animals (wt and db/db) took? - Is angiotensin II the one that you infused? If so, please change to AngII along the manuscript and group of rats - The final scheme could be improved with the localization of proteins in a cell. Diabetes refers to type-II diabetes
· In accordance with the reviewer comment we improved the western blot in figure 1.
· The reviewer is correct in pointing out the early role of HO-1 in the inflammatory response; we have previously shown the effect of CR on cardiac hypertrophy in these mice. Here we wanted to see if the effect of CR is mediated by HO-1.We added a sentence together with the suggested reference (page11, lines 307-311, ref 52)
· The protocol of angiotensin infusion (4 weeks) and CR is described in the methods and is the same as was previously published (ref 30,31, 34) .
· AngII is described as AT along the manuscript.
· The animals not on CR are fed the regular diets (rodent chow, line 101) and are not restricted.
· SnMP did not caused higher mortality.
· The title of final scheme in fig 5 was changed to: “Cellular signaling involved in the development of cardiomyopathy in type-II diabetes and proposed mechanism for the effect of CR on the diabetic heart” as suggested (lines 255-256).
Round 2
Reviewer 1 Report
None of the issues raised were addressed properly
ISSUE1:
"
1. Although the experiments seem well conceived, several confusing data are presented.
First of all, figure 1 shows different mice treatments, whereas in the following table 1 and figure 2 some treatments are missing.
In fig 1 we wanted to showcardiac HO-1 proteins levels in control – WT vs. Diabetic mice. In the other parts of the invivo experiments the role of HO-1 in the diabetic mice alleviating oxidative stress through the activation of HO-1 was emphasized.
"
No data on the other treatments are added, even in a supplementary material
ISSUE2:
"
2. Since these data are HO-1-centered, it would be necessary to measure HO-1 enzymatic activity at least in the in vitro model.
We agree with the reviewer that the enzymatic activity of HO-1 is missing but the fact that in the absence of HO1 either by SIRT or HO-1 inhibition emphasizes crucial role of HO-1 in alleviating oxidative stress.
"
Is not possible to claim a central role of an enzyme without measuring its activity
ISSUE3:
"
3. Figure 5 resumes main findings. In my opinion is not clear if authors proved the interplay between HO-1 and PGC1alpha; moreover since EX527 (SIRT-1 inhibitor) decreases HO-1 level, an arrow from SIRT-1 to HO-1 should be added.
An arrow from SIRT-1 to HO-1 was addedto the scheme in figure 5.
"
This diagram shows involvement in HO-1 in increasing mitochondrial function, that was not measured in this paper. (see ISSUE 4).
In my opinion the data are convincing but conclusions should be more homogeneous with findings.
Author Response
Title: The critical role of Heme Oxygenase 1 in the protective effect of caloric restriction against diabetic cardiomyopathy
Reviewer1:
1. Although the experiments seem well conceived, several confusing data are presented.
First of all, figure 1 shows different mice treatments, whereas in the following table 1 and figure 2 some treatments are missing.
The reviewer is correct in pointing out the differences in the groups between the figures, the full characterization of the model and CR treatment was published in in Cardiovascular Diabetology (ref 31) and therefore not repeated here. We inserted the WT characteristic in table 1 and in the results section (page lines 200-203). Fig 1 exhibits cardiac HO-1 proteins levels in control – WT vs. Diabetic mice. In the other parts of the in-vivo experiments the role of HO-1 in the diabetic mice alleviating oxidative stress through the activation of HO-1 was emphasized.
2. Since these data are HO-1-centered, it would be necessary to measure HO-1 enzymatic activity at least in the in vitro model.
We agree with the reviewer that the enzymatic activity of HO-1 is missing but the fact that in the absence of HO1 either by SIRT or HO-1 inhibition emphasizes crucial role of HO-1 in alleviating oxidative stress. But the fact that SnMP reverses the beneficial effects of increased levels of HO-1 is in this case sufficient. SnMP inhibits HO activity (Drummond et al, ABB 1987) and also increases HO-1 levels (Sardana & Kappas, PNAS 1987); however, the ability to inhibit HO activity takes precedent. Thus, based on these fundamental findings and the effect of SnMP described in this manuscript, it is correct to assume that HO activity is inhibited. We agree that, HO activity should have been measured in vitro. Therefore, we conducted new experiments and measured HO activity (page3 lines 138-143). We add the results of the new measurements as the reviewer requested:” As shown in Figure 3 G, the basal levels of HO activity is inhibited by about 70% in bilirubin formation in presence of SnMP. Increase of glucose levels cause inhibition of HO activity and its further potentiated by SnMP which is clearly observed when glucose reached 33mM”.
3. Figure 5 resumes main findings. In my opinion is not clear if authors proved the interplay between HO-1 and PGC1alpha; moreover since EX527 (SIRT-1 inhibitor) decreases HO-1 level, an arrow from SIRT-1 to HO-1 should be added.
An arrow from SIRT-1 to HO was added to the scheme in figure 5.
4. Finally, it is claimed that the final result is an increase in mitochondrial function; therefore I suggest measuring some specific mitochondrial markers.
Mitochondrial function and an anti-oxidative signaling pathway mediated by SIRT1, and PGC-1α was discussed and measured in the introduction (lines 88-89) and in the discussion (lines 307-317).
5. Please fill-in Authors contributions, Funding, Acknowledgments and Conflict of Interests sections.
We added these details to the end of the article: page 11.
6. Consent for Publication: All authors have seen and approve of the data presented.
Conflict of Interests sections-none
7. The key words were added to page 1.
Reviewer 3 Report
Mostly, authors have not corrected or added the required information
Author Response
Title: The critical role of Heme Oxygenase 1 in the protective effect of caloric restriction against diabetic cardiomyopathy Reviewer3: We thank the reviewer for his comments therefore we corrected the manuscript accordingly: 1.The quality of some blots is very low. Reply: In accordance with the reviewer comment we improved the western blot in figure 1. 2. The data from WT animals should appear also in table 1 Reply: We added the WT data to table 1 3. Similarly, the data of db/db rats should be shown in Figure 2 Reply: We added the functional data from db/db mice to table 1. It should be noted that db/db mice do not develop cardiomyopathy therefore angiotensin infusion was added. We characterized this mouse model and the cardioprotective properties of CR in a previous study: "Regulation of diabetic cardiomyopathy by caloric restriction is mediated by intracellular signaling pathways involving ‘SIRT1 and PGC-1α’." Cardiovascular diabetology 17.1 (2018): 111”. 4. The role of inflammation and HO-1 in cardiac diabetes have been previously described. Thus, this response can be short and early in DCM development, and caloric restriction may be more effective on later responses such as hypertrophy or apoptosis then. Reply: The reviewer is correct in pointing out the early role of HO-1 in the inflammatory response; we have previously shown the effect of CR on cardiac hypertrophy in these mice. In this study we wanted to see if the effect of CR is mediated by HO-1 .We added a sentence together with the suggested reference (page11, lines 332-337, ref 53). 5. Why number of animals is different among the groups? Why is only n=5 in db/db+AngII+CR+SnMP group? Is there any high mortality associated? Reply: Higher mortality was not observed in the SnMP treated group. The other group also served to established the model and to test the effects of CR while the SnMP group served to examine the hypothesis that HO-1 mediated the beneficial effects of CR and As SnMP dramatically reversed CR beneficial effects, a smaller group of mice were tested. 5. Is angiotensin II the one that you infused? If so, please change to AngII along the manuscript-· Reply: The protocol of angiotensin infusion (4 weeks) and CR is described in the methods and is the same as was previously published (ref 30,31, 34). AngII is described as AT along the manuscript. 6. The final scheme could be improved with the localization of proteins in a cell. Diabetes refers to type-II diabetes Reply: The title of final scheme in fig 5 was changed to: “Cellular signaling involved in the development of cardiomyopathy in type-II diabetes and proposed mechanism for the effect of CR on the diabetic heart” as suggested (lines 276-284). 7. English language and style are fine/minor spell check required. reply: The manuscript was reedited
Round 3
Reviewer 1 Report
All issues have been addressed, therefore I consider the paper suitable for publication.